# Exploring the Cholesterol-Modifying Abilities of Lactobacilli Cells in Digestive Models and Dairy Products

**DOI:** 10.3390/microorganisms11061478

**Published:** 2023-06-01

**Authors:** Małgorzata Ziarno, Dorota Zaręba, Iwona Ścibisz, Mariola Kozłowska

**Affiliations:** 1Department of Food Technology and Assessment, Institute of Food Science, Warsaw University of Life Sciences—SGGW (WULS-SGGW), Nowoursynowska 159c St., 02-776 Warsaw, Poland; iwona_scibisz@sggw.edu.pl; 2Professor E. Pijanowski Catering School Complex in Warsaw, 04-110 Warsaw, Poland; dorotazareba@gmail.com; 3Department of Chemistry, Institute of Food Science, Warsaw University of Life Sciences—SGGW (WULS-SGGW), Nowoursynowska 159c St., 02-776 Warsaw, Poland; mariola_kozlowska@sggw.edu.pl

**Keywords:** lactic acid bacteria, gastric juice, intestinal juice, gastrointestinal transit, fatty acid profile, cell survival, health benefits

## Abstract

This study aimed to investigate the ability of lactic acid bacteria to remove cholesterol in simulated gastric and intestinal fluids. The findings showed that the amount of cholesterol removed was dependent on the biomass, viability, and bacterial strain. Some cholesterol binding was stable and not released during gastrointestinal transit. The presence of cholesterol affected the fatty acid profile of bacterial cells, potentially influencing their metabolism and functioning. However, adding cholesterol did not significantly impact the survival of lactic acid bacteria during gastrointestinal transit. Storage time, passage, and bacterial culture type did not show significant effects on cholesterol content in fermented dairy products. Variations in cell survival were observed among lactic acid bacteria strains in simulated gastric and intestinal fluids, depending on the environment. Higher milk protein content was found to be more protective for bacterial cells during gastrointestinal transit than fat content. Future research should aim to better understand the impact of cholesterol on lactic acid bacteria metabolism and identify potential health benefits.

## 1. Introduction

Changes in lifestyle and diet, as well as the development of civilization, have contributed to an increase in so-called “civilization diseases,” also referred to as chronic noncommunicable diseases. Ischemic heart disease, which is also known as coronary artery disease, is an example of a lifestyle disease, with the main cause being the arteriosclerosis of the coronary arteries, leading to their narrowing. Hypercholesterolemia, or elevated LDL cholesterol levels, is one of the risk factors for atherosclerosis. Elevated cholesterol has been identified as a crucial cardiovascular disease risk factor, and even a small reduction in cholesterol levels can reduce the risk of coronary artery disease [1,2,3,4,5]. Cholesterol and oxysterols (the result of cholesterol oxidation) play a significant role in the process of atherosclerotic plaque formation [5]. Atherosclerotic plaques, primarily composed of cholesterol and other substances, are the main cause of narrowing blood vessels and the development of blood clots, which can lead to serious cardiovascular complications such as heart attack and stroke. Oxysterols have the ability to induce inflammation in the blood vessel walls and promote lipid oxidation, contributing to the development and progression of atherosclerosis [5].

Studies have shown that consuming fermented milk products, such as yogurt, can lower cholesterol levels in humans [6,7]. As early as 1974, Mann and Spoerry [8,9] found reduced serum cholesterol levels in African Masai tribe men, attributed to the consumption of large amounts of fermented milk containing wild strains of *Lactobacillus acidophilus*. Lactic acid bacteria (LAB) present in fermented milk are responsible for lowering serum cholesterol levels. Some laboratory studies have also shown that traditional LAB used in the production of cheese, cream, butter, and sour milk can lower cholesterol [1,6,9,10,11,12,13,14,15,16,17]. Much research is available in the literature on cholesterol-lowering properties in laboratory conditions using model media, with most studies involving lactobacilli [10,11,12,13,14,15,16,17]. Other types of bacteria that have shown similar properties are *Bifidobacterium*, *Streptococcus*, *Enterococcus*, *Lactococcus*, and *Leuconostoc* [10,12,18,19,20,21]. This type of research has been conducted since the 1970s [8,17,22,23,24,25,26,27]. However, it is important to note that despite the results of in vitro and in vivo animal and human studies, due to possible methodological and technical errors and lack of reproducibility, the ability of LAB to lower serum cholesterol levels cannot be confirmed or negated unequivocally [28,29,30,31,32,33]. Additionally, the serum cholesterol level not only correlates positively with the amount of cholesterol ingested with food but also depends on the intake of saturated fatty acids and refined carbohydrates. Therefore, there is still no clear confirmation of how LAB have a positive effect on cholesterol levels in humans [34,35,36].

Several hypotheses explain how LAB lower cholesterol levels. Information from the literature suggests that these mechanisms primarily include cholesterol binding, the enzymatic deconjugation of bile salts, the production of exopolysaccharides, and the synthesis of short-chain fatty acids [12,18,20,21,34,35,37,38,39,40,41,42,43,44,45,46]. It has been found that LAB can bind cholesterol either through adhesion or assimilation [13,18,20,21,37,47,48,49,50]. Studies have also shown that LAB can hydrolyze (deconjugate) bile salts and bind to cholesterol molecules, thereby lowering cholesterol levels in the system [7,12,15,36,42,44,51,52,53,54,55,56,57]. Although the exopolysaccharides produced by many species of LAB can bind cholesterol molecules, this mechanism is still among the studied hypotheses [34,35,40,45,58,59]. In the context of research on the impact of LAB on cholesterol levels, it is essential to consider the relationship between reducing cholesterol levels and reducing the risk of atherosclerosis and cardiovascular diseases [46,50]. This information will help readers understand the significance of studying the effects of LAB on cholesterol levels and their potential therapeutic value in the prevention of cardiovascular diseases.

The adhesion and assimilation of cholesterol by LAB can occur simultaneously when the bacterial cells are alive and biologically active. However, it is also possible that cholesterol only undergoes adhesion when the bacterial cells are dead. Understanding the factors that affect cholesterol binding across the cell wall or membrane will help determine which cholesterol-binding mechanism dominates under specific environmental conditions. The removal of cholesterol from the culture medium may indicate the possibility of cholesterol binding in the food product during fermentation or refrigerated storage. However, bound cholesterol may still be bioavailable for the human body. Therefore, separate experiments on the binding, removal, and persistence of cholesterol in the human gastrointestinal tract should be performed to determine if the release of bound cholesterol occurs under these conditions. Such research can be easily conducted in vitro using classical culture media or conditions that simulate the human digestive system. These findings suggest that the positive effects of LAB on cholesterol levels can also occur under in vivo conditions [29,30,31,32,33].

The aim of this study was to investigate the ability of LAB cells to remove cholesterol from different in vitro conditions and to examine the impact of selected factors on the quantity of cholesterol removed and the release of previously bound cholesterol.

## 2. Materials and Methods

### 2.1. Materials

The biological material used in this study consisted of four lactobacilli strains, namely *Lactobacillus delbrueckii* subsp. *bulgaricus* ATCC 11842, *Lactobacillus helveticus* LH-B01, *Lactobacillus delbrueckii* subsp. *lactis* ATCC 4797, and *Lactobacillus acidophilus* La-5, which were deposited in the institute’s collection. Before use, the strains were reactivated and propagated in MRS broth aliquots (at 37 ± 1 °C for 18 h). Their biomass was then prepared at different cell concentrations: 10× concentrated biomass (10×), nonconcentrated biomass (1×), and 10× diluted biomass (0.1×). To obtain 10-fold concentrated biomass, a 17 h broth culture was centrifuged in an ultracentrifuge (model type 317a, Mechanika Precyzyjna, Warsaw, Poland) for 7 min at 12,000× *g* at 4 ± 1 °C. In contrast, 10-fold diluted biomass was obtained by diluting the initial 17 h broth culture (at 1-fold biomass concentration) 10 fold, followed by mixing on a micro shaker. The number of viable bacterial cells in the culture broth was determined using the deep plate method (MRS agar, 37 ± 1 °C, 72 h, anaerobic conditions). Dead-cell biomass was obtained by sterilizing (121 °C/15 min) suspensions of live bacterial cells in the culture broth.

### 2.2. Media and Other Reagents

The MRS broth (basic) was prepared in the appropriate amount according to the manufacturer’s instructions (Merck, Darmstadt, Germany). A model MRS broth solution with added cholesterol was also prepared by measuring a suitable amount of dissolved sterile cholesterol solution into the basic sterile MRS broth to achieve the desired final concentration of cholesterol in the broth. MRS agar, used to determine lactobacilli cell counts, was prepared by dissolving a powdered MRS agar medium (Merck) in distilled water, following the manufacturer’s instructions.

To prepare the cholesterol solution, powdered cholesterol with >99% chemical purity (Sigma-Aldrich, St. Louis, MO, USA) was dissolved in a mixture of 99% ethanol and Tween 80 (Merck) in a 3:1 ratio. The dissolution was performed by placing the bottle containing the prepared mixture of ethanol, Tween 80, and cholesterol in a hot water bath (approximately 95 ± 1 °C). The cholesterol concentration in the resulting solution was about 40 g/L. The Cholesterol RTU^®^ enzymatic cholesterol kit (BioMérieux Polska, Warsaw, Poland) was used to determine the cholesterol concentrations in the culture samples.

The model gastric juice used in this study was carried out following the publication of Clavel et al. [60]. To prepare the model gastric juice, 4.8 g NaCl (POCH Polish Chemicals Reagents, Gliwice, Poland), 1.56 g NaHCO_3_ (POCH), 2.2 g KCl (POCH), and 0.22 g CaCl_2_ (POCH) were dissolved in 1000 mL distilled water. The resulting solution was then adjusted to pH 2.4 ± 0.2 using a pH meter (model LPH330T, Grosseron, Tacussel Electronique, Coueron, France) and a 1 M HCl (POCH) solution. The model gastric juice was sterilized in portions of 50 mL in an autoclave at 121 ± 1 °C for 15 min. In the experiments, a freeze-dried preparation of pepsin (Sigma-Aldrich) with a potency of 3200–4500 units/1 mg protein was used. Pepsin was added to the model gastric juice immediately before the experiments (2 mg crystalline pepsin to 50 mL model gastric juice).

The intestinal juice model used in this study was prepared following the publication of Marteau et al. [61], with some modifications. First, 2.5 g NaCl (POCH), 0.3 g KCl (POCH), 0.015 g CaCl_2_ (POCH), and 8.5 g bovine bile (Merck) were dissolved in 500 mL of the previously prepared 1 M NaHCO_3_ solution. The pH of the solution after sterilization was determined to be 7.0 ± 0.2 using a pH meter (model pHT 003, Eon Trading LLG, USA) and 1 M NaOH (POCH). The solution was then autoclaved at 121 ± 1 °C for 15 min. A sterile solution of intestinal enzymes (750 mg pancreatin from bovine pancreas with an activity of 50,000 FIP units lipase, 40,000 FIP units amylase, and 3000 FIP units protease) was added to 160 mL portions of the intestinal juice model immediately before use.

All the chemicals used in this study were of analytical grade or HPLC grades and were purchased from Merck or Sigma-Aldrich Co. The standards used in this study to identify fatty acids included oleic acid (Sigma-Aldrich), anteiso12-methyltetradecanoic acid (Sigma-Aldrich), 2-hydroxytetradecanoic acid (Sigma-Aldrich), nonadecanoic acid (Sigma-Aldrich), bacterial acid methyl esters (BAME; Sigma-Aldrich), GLC-674, and GLC-617 (Nu-Chek-Prep., Elysian, MI, USA). Additionally, the isomers of methyl esters of linoleic acid 18:2 (cis-9,trans-11 and trans-10,cis-12; Nu-Chek Prep., Elysian, MN, USA) were also utilized. In cases where other fatty acids were identified, comparisons were made with literature data [62,63,64]. For the determination of cholesterol in samples, 5α-cholesterol (Sigma-Aldrich) was used as an internal standard.

### 2.3. Methods

#### 2.3.1. Testing the Ability of Lactobacilli Cells to Remove Cholesterol under Gastric Juice and Intestinal Juice Conditions

To determine the degree of cholesterol removal, the biomass compounds of the strains were incubated in juice models containing the added cholesterol solution. First, the biomass compounds were incubated in gastric juice (1:1) at 37 °C for 3 h, followed by incubation in intestinal juice (1:1) at 37 °C for 5 h. The cholesterol concentration in the culture liquid was determined using an enzymatic cholesterol determination kit at the beginning and end of each incubation procedure. Separate experiments were performed using biomass from live and thermally inactivated bacterial cells.

#### 2.3.2. Testing the Ability of Lactobacilli Cells to Release Previously Bound Cholesterol under Gastric Juice and Intestinal Juice Conditions

To allow for initial cholesterol binding, culture cell biomass samples were cultured in MRS broth containing the added cholesterol at 37 °C for 24 h. After culture, the cholesterol concentration in the postculture liquid was determined using an enzymatic cholesterol determination kit, and the biomass samples were centrifuged in an ultracentrifuge (model type 317a) for 7 min at 12,000× *g* at 4 ± 1 °C. The resulting biomass samples were then incubated in juice models containing no added cholesterol solution. First, the biomass samples were incubated in gastric juice (1:1) at 37 °C for 3 h, followed by incubation in intestinal juice (1:1) at 37 °C for 5 h. The cholesterol concentration in the culture liquid was determined using an enzymatic cholesterol determination kit at the beginning and end of each incubation procedure. Separate experiments were performed using biomass from live and thermally inactivated bacterial cells.

#### 2.3.3. Testing the Ability of Lactobacilli Cells to Remove Cholesterol from Dairy Products under Gastric and Intestinal Juice Conditions

Model dairy products were prepared using fresh skim milk (0.05% fat) with the addition of cream (12% fat) or skimmed milk powder to adjust the fat and dry-matter content (DM). Five variants of model dairy products were prepared: (1) fresh milk with 0.05% fat and 12% DM; (2) fermented milk with 0.05% fat and 12% DM; (3) fermented milk with 0.05% fat and 20% DM; (4) fermented milk with 12% fat and 12% DM; and (5) fermented milk with 12% fat and 20% DM. The fermentation of samples of selected variants of model dairy products was carried out at 37 ± 1 °C for 5 h. The finished products were then refrigerated at 6 ± 1 °C and stored for 4 weeks. After this period, samples of the model dairy products were stored in juice models without the added cholesterol solution, first in gastric juice (1:1, at 37 °C for 3 h) and then in intestinal juice (1:1, 37 °C for 5 h). Simultaneously, the test strains were cultured in MRS broth with and without the addition of cholesterol solution. After the completion of fermentation and cooling of the samples, after 1 and 4 weeks of refrigerated storage, and after the end of storage under the conditions of juice models (gastric and intestinal juice), the cholesterol content in the samples was determined using a gas chromatograph model GC-MS QP 2010 (Shimadzu, SHIM-POL A.M. Borzymowski, Warsaw, Poland). The GCMS Solution v.2.50 software (Shimadzu, SHIM-POL A.M. Borzymowski, Warsaw, Poland) was used for analysis of the results. The final cholesterol content results were converted to the starting amount of the model dairy products, considering the dilution levels of the samples at each stage of the experiments. The number of viable bacterial cells in the samples was determined using the plate depth method (MRS agar, 37 ± 1 °C, 72 h, anaerobic conditions).

#### 2.3.4. Determination of Cholesterol Content in Model Dairy Products Using Gas Chromatography Technology Combined with Mass Spectrometry

Fat extraction from the samples was carried out through direct saponification [65]. In brief, 0.2 g of the sample was weighed into a tube, and 5 mL of 0.5 M methanolic KOH solution was added. The tube was then capped and vortexed for 15 s. Subsequently, the tube was heated in an 80 °C water bath for 15 min with vortexing every 5 min for 10 s. After heating, the sample was cooled, and 1 mL of water and 5 mL of n-hexane (Merck, Darmstadt, Germany) were added. Finally, the sample was vigorously vortexed for 1 min and centrifuged at 2000× *g* for 1 min. An aliquot of the upper phase was then transferred to an autosampler vial and analyzed using GC-MS QP 2010. The gas chromatograph operated under the following conditions: a DB-type column measuring 30 m/0.25 m/0.25 mm; the initial temperature of 50 °C with an isotherm of 2 min; temperature increased by 25 °C/min to 270 °C with an isotherm of 1 min; and temperature ramp of 5 °C/min to an end temperature of 320 °C with an isotherm of 10 min. The analysis duration was 31.80 min. The retention time for cholesterol was about 18.2 min, and for 5-cholestane, it was 15.7 min. The dispenser temperature was 260 °C, with injection in split mode, a split ratio of 1:30, column gas flow of 0.98 mL/min, and a pressure of 52.3 kPa. The mass spectrometer operated under the following conditions: ion-source temperature of 220 °C, the temperature of the connecting line between GC and MS being 240 °C, a voltage of 1.13 kV at the detector, the ionization energy of 70 eV, and a quadrupole filter sweep range of 100–600 *m*/*z*.

#### 2.3.5. Determination of the Fatty Acid Profile of Lactobacilli Cells in MRS Broth with and without the Addition of Cholesterol Solution

The fatty acids were extracted using the method described in the references [66,67,68] and the gas chromatography coupled to a mass spectrometer was used for the chromatographic separation of fatty acid methyl esters (GC-MS QP 2010), with a 007-23-30-0.2F polar column (30 m/0.25 mm/0.20 m; Quadrex). The sample was injected at a split ratio of 1:25 with a dispenser temperature of 230 °C. The chromatographic separation was carried out using the following conditions: an initial column temperature of 60 °C with a 2 min isotherm, a temperature ramp of 4 °C/min to 220 °C, and a 10 min isotherm. The carrier gas used was helium, with a flow rate of 0.37 mL/min. The detector conditions used were as follows: ion-source temperature of 200 °C, the temperature of the line connecting GC to MS of 220 °C, detector voltage of 1.45 kV, and quadrupole filter sweep range of 50–400 *m*/*z*.

### 2.4. Statistical Analysis

Each experiment was carried out in three independent replicates (*n* = 3), and each replicate was measured twice. The data were subjected to a two-way analysis of variance (ANOVA), and the mean differences between the statistical groups were tested at a significance level of α = 0.05 using Tukey’s test. Multivariate analysis was employed to describe the relationship of multiple variables for each sample at a significance level of α = 0.05. The Statgraphics Centurion XVII program (Statgraphics Technologies, Inc., The Plains, VA, USA) was used for statistical analyses.

## 3. Results and Discussion

### 3.1. Testing the Ability of Lactobacilli Cells to Remove Cholesterol under Gastric Juice and Intestinal Juice Conditions

The experiments were conducted under in vitro conditions, and the results are presented in Figure 1a,c,e,g. The chosen experimental setup facilitated the simultaneous examination of cholesterol removal during the transit of lactobacilli cells through a segment of gastric and intestinal juice models. It was assumed that LAB do not metabolize cholesterol, and therefore, the loss of cholesterol from the culture broth should be directly proportional to the amount of cell biomass that binds and removes cholesterol. Consequently, the impact of bacterial biomass concentration and viability on the observed phenomena of cholesterol removal and release during the passage of lactobacilli cells through a section of gastric and intestinal juice models was investigated.

The initial concentration of cholesterol in the cultures of the lactobacilli strains tested under gastric juice and intestinal juice conditions was 0.702 g/dm³. The findings revealed that the removal of cholesterol by LAB cells under gastric juice and intestinal juice conditions was significantly dependent on the biomass amount and viability of the bacterial cells and also varied depending on the bacterial strain under study. Among the tested strains, the viable bacterial cells of the *L. helveticus* strain LH-B01 exhibited the highest level of cholesterol removal under the gastric juice condition, increasing the concentration 10 fold (average 0.043 g/dm³, Figure 1g).

Liong and Shah [41] noted that the number of cells significantly impacts the differences in the amount of cholesterol bound by LAB, whereas the growth dynamics of individual strains determine the amount of cell biomass and differences in experimental results. However, the absence of sufficient literature data in this area precludes a more detailed discussion of the results obtained in this study. Nonetheless, our study revealed that the initial concentration of bacterial cell biomass had a positive effect on the amount of cholesterol removed from the culture medium. However, it was expected that as the concentration of cell biomass decreased by a factor of 10, the amount of cholesterol removed from the culture broth would also decrease proportionally, by 10 fold). Interestingly, little difference was observed in the amount of cholesterol removed using biomass samples with varying concentrations of living cells used. These results could provide clues to explain the hypocholesterolemic effect of products containing LAB, an area that has been widely investigated in the literature but with contradicting findings [8,9,22,69,70,71,72].

For bacterial cells to take up cholesterol molecules, their high biological activity is necessary since, as demonstrated by Hosono and Tono-Oka [18], this phenomenon occurs most intensely during the logarithmic growth phase in lactic acid streptococci. The physical binding of cholesterol through the cell wall does not require cellular activity but only requires a sufficiently long contact time between the cells and cholesterol molecules. Additionally, the same researchers found that not only living but also dead (autoclaved) cells of the tested strain could bind cholesterol. The fact that cholesterol removal occurs even when bacterial cells are dead confirms that the physical binding of cholesterol molecules through the cell wall (adhesion) is one of the mechanisms of cholesterol removal by LAB cells [1,41,73,74,75]. These observations align with the experimental results presented in this study (Figure 1a,c,e,g).

### 3.2. Testing the Ability of Lactobacilli Cells to Release Previously Bound Cholesterol under Gastric Juice and Intestinal Juice Conditions

In addition to the phenomenon of cholesterol binding and removal, it is also crucial to study whether bound cholesterol remains bioavailable to the human body. The experiments carried out in this part of the study, the results of which are presented in this section, addressed this issue. The objective of the experiments was to investigate whether the cholesterol previously bound by LAB cells is released under gastric juice or intestinal juice conditions. The results of the experiments are illustrated in Figure 1b,d,f,h. Initially, the living- and dead-cell biomass samples of the tested lactobacilli were cultivated in MRS broth with added cholesterol (average concentration of 0.647 g/dm^3^); after 24 h, the cell biomass samples were transferred to a gastric juice model for 3 h, and thereafter to an intestinal juice model for 5 h.

The results of the experiments suggest that cholesterol previously removed and bound by cells of monocultures of LAB can be released. The experiments on the release of cholesterol by lactobacilli cells during their passage through a section of gastric and intestinal juice models revealed significant differences between the biomass of living and dead bacterial cells. Among the living lactobacilli cells, the strain *L. acidophilus* La-5 removed the most cholesterol from the initial amount of cholesterol in the MRS broth (0.677 g/dm³), with an average of 0.104 g/dm³ (Figure 1g). However, during the passage through a section of gastric and intestinal juice models, the same strain released an average of 0.082 g/dm³ of cholesterol, which was significantly less than what was removed from the MRS broth. In contrast, the dead-cell biomass of the *L. acidophilus* La-5 strain removed the most cholesterol from the MRS broth among the lactobacilli strains tested, with an average of 0.056 g/dm³, and released an average of 0.044 g/dm³ of cholesterol during the passage through a section of gastric and intestinal juice models. Statistical analyses showed that in this case, the amount of cholesterol released also depended on the viability of the bacterial cells. The dead-cell biomass of the lactobacilli strains released less bound cholesterol than the living-cell biomass, but they also removed less bound cholesterol from the culture medium earlier.

The results suggest that a small portion of cholesterol bound by LAB cells is so firmly bound that it is not released during the passage of lactobacilli cells through the section of gastric and intestinal juice models. It can be assumed that the cholesterol bound in this manner is not bioavailable to the human body [1]. This finding is consistent with those of Albano et al. [1], Lee et al. [76], and Miremadi et al. [77] on structural changes in the bacterial cell wall. The results obtained in this study can be related to studies on the binding of aflatoxin B1 by LAB cells [76,78]. El-Nezami et al. [78] observed that the removal of aflatoxin B1 (AFB1) from the culture medium using selected cultures of LAB depended on their population and culture temperature. Lee et al. [76] also investigated the phenomenon of binding and release of AFB1 bound using the living and dead cells of *L. rhamnosus* GG and *L. rhamnosus* LC 705 and found similar relationships to those observed in this study regarding the binding and removal of cholesterol using LAB cells. Lee et al. [76] also concluded that the thermal killing of the bacteria altered the bacterial cell surface and exposed additional binding sites for AFB1.

Most studies on the effect of LAB on cholesterol have been conducted using classic culture broths, sometimes with the addition of bile salts [8,15,27,46,77,79]. However, there are no studies available in the literature on the effect of LAB cells on cholesterol binding under the conditions of an intestinal juice model. It is important to note that the intestinal juice model used in our experiments contained bovine bile with both conjugated and deconjugated bile salts, meaning that the bile salt hydrolase activity produced by most intestinal strains of LAB was not required to precipitate cholesterol with free bile acids [27,36,42,43,44,51,54,55,80]. The bile salt that is not conjugated exhibits diminished solubility, lesser absorption efficiency, and reduced effectiveness in emulsifying fat and facilitating the absorption of cholesterol [46]. The coprecipitates of cholesterol with bile acids are known to form at low pH, below 5.5 [14,15,31,51,81]. Although the intestinal juice model used in the present study had a pH above 7.0, it is possible that the pH dropped to a level sufficient for the coprecipitation of cholesterol with free bile acids due to the addition of viable bacterial cells. This is possible in stationary cultures and is confirmed in the literature cited [15,27,36,42,43,44,52]. Such coprecipitates would rapidly dissolve under in vivo conditions if the pH rose above 5.5 [14,16,34,42], as the bile secreted by the liver travels to the duodenum, where it neutralizes the acidic chyme leaving the stomach, and the pH in the small intestine is above 6.0. Therefore, the hypocholesterolemic effect caused by the coprecipitation of cholesterol with deconjugated bile acids is unlikely to occur under in vivo conditions. However, this does not mean that lactobacilli are not beneficial in modifying other physiological parameters indicative of markers of metabolic syndrome, including obesity, hyperlipidemia, hyperglycemia, and insulin resistance, under in vivo conditions [29,30,31,82]. The passage of lactobacilli cells through the section of gastric and intestinal juice models may have implications for the hypocholesterolemic effect observed in this study. Therefore, when considering the passage of lactobacilli cells through the section of gastric and intestinal juice models, it is important to acknowledge the potential for pH changes and the coprecipitation of cholesterol with bile acids. However, in a physiological setting, the dissolution of such coprecipitates is likely, limiting the hypocholesterolemic effect of lactobacilli observed in vitro. The hypocholesterolemic effect of some probiotics characterized by high BSH enzyme activity in vitro has also been confirmed in vivo in both humans and animals [8,32,33,46,83].

### 3.3. Testing the Ability of Lactobacilli Cells to Remove Cholesterol from Dairy Product Models under Gastric Juice and Intestinal Juice Conditions

The previous experiments conducted in this study demonstrated that lactobacilli cells can effectively remove and bind cholesterol from both culture broth and digestive juice models. This process is contingent upon the number of bacterial cells present and their viability. It is worth noting that the consumption of fermented milk products involves the ingestion of a significant amount of LAB cell biomass.

The objective of the current experiment was to assess the cholesterol-removing ability of lactobacilli in dairy product models that vary in DM, fat content (0, 12, and 20%), and acidity (fresh milk and fermented milk). The products were subjected to 4 weeks of refrigerated storage and subpassage conditions in a section of gastric and intestinal juice models. The results of this experiment are presented in
Table 1.

The ability of specific lactobacilli cultures to lower cholesterol in culture broth has not been shown to have a direct correlation with the reduction in cholesterol levels in fermented milk products. In this study, it was found that the cholesterol content of the dairy product models was only significant with the original total fat content of the samples. Neither the length of cold storage nor the passage of gastric and intestinal juice models had any significant effect on the reduction in cholesterol levels. It is worth noting that the type of lactobacilli culture used did not have a significant impact on the cholesterol content. This is to be expected since LAB cells are not capable of metabolizing cholesterol; instead, they can only bind to it through the cell wall and/or incorporate it into the cell wall or membrane simultaneously.

Although the exact mechanism of cholesterol binding by bacterial cells remains unclear, it is expected that the process of lipid extraction from a sample would extract all the cholesterol present in the sample. However, the results obtained in this portion of the study have yet to be confirmed in the literature. Aloglu and Öner [79] conducted research on *Lactobacillus* cultures to assess their cholesterol-removing abilities in culture media containing added bile salts and cholesterol (at 0.150 g/dm³) as well as cream. They observed a reduction in cholesterol levels using bacterial cells in both the culture broth and cream. The percentage of cholesterol removed from the cream ranged from 20.6% to 59.8% of its initial level, whereas the same cultures removed 12.1–47.5% of the initial cholesterol content from the culture medium.

### 3.4. Survival of Lactic Acid Bacteria Cells under Digestive System Conditions

The experiments discussed above suggest that lactobacilli cells have the ability to remove cholesterol from the culture medium and gastric and intestinal juice models, as well as release already-bound cholesterol. It was demonstrated that the amount of cholesterol removed or released is dependent on the quantity and viability of bacterial cell biomass. Therefore, this study aimed to investigate whether lactobacilli cells, suspended in model dairy products with varying DM, fat content (0, 12, and 20%), and acidity (fresh milk and fermented milk), could survive in an environment that simulates the conditions of the stomach or intestines. For comparison, the same conditions were applied to biomass samples of the tested lactobacilli, suspended in MRS broth with or without the addition of cholesterol. The results of these experiments are presented in Figure 2a–d.

The data presented in this study suggest that the lactobacilli cells used in these experiments were as active as in previous studies and exhibited good survival rates under refrigerated storage conditions of the dairy samples model. Before the passage of gastric and intestinal juice models, the number of viable lactobacilli cells in all milk product model samples tested ranged from 7.2 log(CFU/mL) to 8.3 log(CFU/mL), regardless of the culture used. However, a 3 h incubation procedure under gastric juice conditions resulted in a significant reduction in the population of viable lactobacilli cells, depending on the type of sample in which the cells were suspended and the strain tested. *L. helveticus* strain LH-B01 suspended in MRS broth, fresh milk samples, or 12% fat fermented milk samples exhibited the lowest survival rates under these conditions (Figure 2b). For comparison, a 3 h incubation procedure in the gastric juice model most significantly reduced the population of live *L. acidophilus* La-5 cells (Figure 2d). In addition, a 5 h incubation process in the intestinal juice model resulted in the elimination of live lactobacilli cells suspended in MRS broth to below the limit of detection (<0.1 log(CFU/mL)), regardless of the strain tested. However, it is noteworthy that there was no significant effect of cholesterol supplementation on the survival of lactobacilli under gastric or intestinal juice conditions. When lactobacilli cells were suspended in dairy product models, higher survival rates were observed, particularly for *L. acidophilus* La-5 cells (Figure 2d).

Research suggests that nonprobiotic bacterial strains exhibit lower cell survival rates than probiotic strains under gastric juice conditions [39,84]. Vinderola and Reinheimer [84] conducted a study on the tolerance of probiotic strains of LAB and bifidobacteria to gastric juice conditions and found that *L. acidophilus* cells were the most resistant to low pH levels. The good tolerance of bacterial cells to digestive juice conditions can be explained by the presence of these bacteria in the digestive tracts of humans or animals. Bacteria that are not natural gut flora lack natural resistance to intestinal juice conditions [85]. However, this is not conclusive since Elli et al. [86] demonstrated that, under in vivo conditions, some *Streptococcus thermophilus* can survive in the human gastrointestinal tract, despite not being a typical intestinal flora. In their study, a group of 20 volunteers were fed yogurt containing live cells of LAB. Researchers found live cells of *S. thermophilus* and *L. delbrueckii* subsp. *bulgaricus* in the terminal gastrointestinal tract and feces. Therefore, it can be assumed that the survival of lactobacilli cells in the human gastrointestinal tract is dependent on the individual properties of the strains of these bacteria and is not a property of the entire species, which has also been confirmed by several other researchers [20,21,33,83].

It is important to note that the results of this study demonstrated statistical differences in cell survival rates among individual lactobacilli strains under gastric or intestinal juice conditions, depending on the environment in which the bacteria entered the tested system (Figure 3). The analysis conducted in this study on the relationship between pH, fat content, DM content, and viability of the tested lactobacilli strains (*L. delbrueckii* subsp. *bulgaricus* ATCC 11842, *L. helveticus* LH-B01, *L. delbrueckii* subsp. *lactis* ATCC 4797, and *L. acidophilus* La-5) revealed that, after the incubation of the samples under gastric juice conditions for 3 h, the population of living lactobacilli cells was more strongly determined by the DM content of the samples (correlation coefficients of 0.89, 0.77, and 0.69 for *L. delbrueckii* subsp. *bulgaricus* ATCC 11842, *L. acidophilus* La-5, and *L. delbrueckii* subsp. *lactis* ATCC 4797, respectively). In other cases, the DM content did not significantly affect the cells more than the fat content of the samples (a correlation coefficient of 0.44 for *L. delbrueckii* subsp. *bulgaricus* ATCC 11842; in other cases, the factor was not significant). In contrast, the pH of the samples was an inhibiting factor for the survival of cells of specific lactobacilli strains under gastric juice conditions (a correlation coefficient of −0.59 for the strains *L. delbrueckii* subsp. *bulgaricus* ATCC 11842 and *L. acidophilus* La-5; in other cases, this factor was not significant). Similar results were obtained after statistical analysis of the results obtained after incubating the samples under intestinal juice conditions for 5 h. The analysis of the relationship between pH, fat content, DM content, and viability of the tested lactobacilli strains performed in this case showed that the population of all the tested lactobacilli strains was more strongly determined by the DM content of the samples (correlation coefficients of 0.97, 0.94, 0.93, and 0.91 for *L. delbrueckii* subsp. *bulgaricus* ATCC 11842, *L. delbrueckii* subsp. *lactis* ATCC 4797, *L. helveticus* LH-B01, and *L. acidophilus* La-5, respectively) than the fat content of the samples (correlation coefficients of 0.60, 0.48, and 0.47 for *L. delbrueckii* subsp. *lactis* ATCC 4797, *L. acidophilus* La-5, and *L. delbrueckii* subsp. *bulgaricus* ATCC 11842, respectively). In other cases, this factor did not significantly affect the cells. In addition, the pH of the samples was an inhibiting factor for the survival of cells of each lactobacilli strain under gastric juice conditions (correlation coefficients of −0.45, −0.50, −0.50, and −0.52 for *L. delbrueckii* subsp. *bulgaricus* ATCC 11842, *L. acidophilus* La-5, *L. delbrueckii* subsp. *lactis* ATCC 4797, and *L. helveticus* LH-B01, respectively). This may indicate that the dry weight of the products (in this case, the significant content of milk-derived proteins) has a more protective effect on lactobacilli bacterial cells than the fat content during passage through the stomach and intestines simulations.

### 3.5. Determination of the Fatty Acid Profile of Lactobacilli Cells in MRS Broth with and without the Addition of Cholesterol Solution

The survival of LAB cells under digestive system conditions is influenced by factors such as the probiotic nature of the strains, their inherent resistance to low pH levels, and individual strain properties rather than being solely determined by their classification as natural gut flora. Various factors have been identified that can protect bacterial cells from the harsh conditions of the gastrointestinal tract, including acidity, organic acids, bile salts, nutrients, and the length of time the bacteria spend in the gut, as well as their initial concentration. Interestingly, cholesterol has also been shown to enhance the survival of bacteria in the stomach and intestines [52]. This is thought to be due to cholesterol’s ability to make LAB more resistant to lysis and alter the composition and function of the cell wall and membrane, thus changing their tolerance to environmental factors [34,52]. Therefore, in this study, the effect of cholesterol on the fatty acid profile of lactobacilli cells in MRS broth, with and without the addition of cholesterol solution, was also investigated. The results of these experiments are presented in Table 2, while the composition of the external standard (BAME) and the identification parameters for each fatty acid are given in Table 3.

The fatty acid profile of the lactobacilli studied was analyzed using the chromatographic separation of the extracted fatty acids from the bacterial biomass, identifying a total of 29 different fatty acids. Among these, six saturated fatty acids, two single-branched fatty acids with *iso* and *anteiso* structures, one *hydroxy* fatty acid, nine monounsaturated fatty acids with a single double bond, two polyunsaturated fatty acids with multiple double bonds, six conjugated fatty acids, and three cyclic fatty acids were identified. The most predominant fatty acids found in the bacterial biomass were C14:0; C16:0; C16:1, *cis*-9; C18:0; C18:1, *cis*-9; C18:1, *cis*-11; *cycC19*:0, *cis*-9,10; and *cycC19*:0, *cis*-10,11. Significant differences were observed among the fatty acid profiles of the different lactobacilli strains studied. The largest differences in fatty acid profiles were observed for 12 fatty acids, including C10:0, 15:0, *iso*; C16:1, *trans*-9; C16:1, *cis*-9; *cycC17*:0, *cis*-9,10; C18:1, *trans*-6; C18:1, *cis*-6; C18:1, *cis*-9; C18:2, *trans*-9,*trans*-12; C18:2, *cis*-9,*cis*-12; *cycC19*:0, *cis*-9,10; and *cycC19*:0, *cis*-10,11. Interestingly, two common monounsaturated fatty acids, C16:1, *cis*-9; and C18:1, *cis*-9, were identified as substrates for cell-membrane synthesis and can enhance membrane flexibility and fluidity, thereby preventing cell damage and lysis. In contrast, linoleic acid isomers such as C18:2, *cis*-9,*cis*-12 as well as C18:2, *trans*-9,*trans*-12 were found to affect metabolic functions of lactobacilli, such as lipid synthesis and fatty acid biosynthesis, and reduce resistance to environmental stresses such as high temperature or high salt concentrations.

The results presented here demonstrate that the presence of cholesterol in MRS broth can impact the fatty acid profile of lactobacilli cells, as previously suggested by Miremadi et al. [77]. The addition of cholesterol in MRS broth induced significant changes in the fatty acid profile of cells from the tested lactobacilli strains. The incubation of *L. delbrueckii* subsp. *bulgaricus* ATCC 11842 and *L. delbrueckii* subsp. *lactis* ATCC 4797 strains in MRS broth with the added cholesterol resulted in an increased proportion of C18:2, *cis*-9,*cis*-12 (linoleic acid) in the fatty acid pool. In the cells of *L. helveticus* LH-B01, a significant reduction in the level of C16:1, *cis*-9 (palmitoleic acid) was observed, while in the cells of the *L. acidophilus* La-5 strain, there was a significant reduction in the level of two fatty acids: C18:1, *cis*-9 (oleic acid) and *cycC19*:0, *cis*-9,10 (dihydrosterculic acid) in the fatty acid pool. These observed changes in the fatty acid profile can influence the structure and properties of the cell membrane, which can in turn affect the metabolism and function of these bacteria. For instance, increasing the level of linoleic acid (C18:2, *cis*-9,*cis*-12) can elevate cell-membrane fluidity and thus enhance the adaptability of bacteria to changing environmental conditions. Conversely, reducing the level of oleic acid (C18:1, *cis*-9) can increase the acidity of the cell membrane, which can affect the enzymatic and transport properties of cells. Liong and Shah [42] also examined the effect of cholesterol on the fatty acid profile of *L. acidophilus* bacterial cells and discovered that the strains grown without the addition of cholesterol had a higher proportion of unsaturated acids (oleic and linolenic acid) than samples with the addition of cholesterol solution. Changes in the fatty acid profile of bacterial cells also suggest the incorporation of cholesterol into the cell membrane [75]. Boudreau and Arul [87] also found that the presence of cholesterol increases certain saturated fatty acids compared with samples without the addition of this ingredient. However, it should be noted that the effects of individual fatty acids on the cell membrane and bacterial function are complex and depend on several factors such as the type of fatty acid, its concentration, and its relationship to other fatty acids. Nevertheless, it is worth mentioning that these results are based on in vitro studies, and their relevance to the physiology of lactobacilli in the human or animal body is not clear. Moreover, the effect of cholesterol on the fatty acid profile of lactobacilli cells may depend on several factors such as incubation time, cholesterol concentration, and environmental composition.

The cytoplasmic membrane is essential for the survival of bacterial cells, and the biosynthesis of cell-membrane components is a crucial aspect of bacterial physiology. The synthesis of fatty acids must be controlled, as it coordinates the production of membrane lipids [88,89,90]. The basic type II biosynthetic pathway of LAB fatty acids is a repetitive cycle of condensation, reduction, dehydration, and reduction in C–C bonds. Lipid components act as a barrier, controlling the permeability of the membrane and contributing to the asymmetry of the lipid membrane, which is necessary for the survival and functioning of bacterial cells [27,76]. The fatty acid profile of the LAB cell membrane depends on various factors such as temperature, pH, oxygen, growth phase, the composition of the medium, and salt concentration [64,91]. Corcoran et al. [92] demonstrated that C18:1, *cis*-9 (oleic acid) and C18:1, *cis*-11 (*cis*-vaccenic acid) have a protective effect on *L. rhamnosus* GG cells suspended in artificial gastric juice. They showed that only in the case of these two fatty acids, the number of bacterial cells was greater than the number of cells in the control sample. Other tested fatty acids, including C18:0 (stearic acid); C18:1, *trans*-9 (elaidic acid); C18:2, *cis*-9,*cis*-12 (linoleic acid); C18:2, *cis*-9,*trans*-11 (*cis*-9,*trans*-11-octadecadienoic acid); and C18:2, *trans*-10,*cis*-12 (*trans*-10,*cis*-12-octadecadienoic acid), caused a decrease in the number of viable cells to a level of 3.9 log(CFU/mL) in the case of *trans*-10,*cis*-12-octadecadienoic acid and 4.8 log(CFU/mL) for stearic and elaidic acids. This phenomenon can be explained by the fact that oleic and *cis*-vaccenic acids are the substrates for the synthesis of fatty acids required for cell survival and the modification of cell-membrane fluidity. LAB are equipped with mechanisms to convert these fatty acids into their cyclic, polyunsaturated, or conjugated forms. Similar observations were made by Taranto et al. [52], who studied the effects of bile acids and cholesterol on the fatty acid profile of cells of *L. reuteri* CRL 1098. Their analysis of the fatty acid profile showed that 50% of the fatty acids present were C16:0, C18:1, C18:2, C18:0, and C19:0, present in varying proportions depending on the medium used. Taranto et al. [49] also observed a similarity between the acid profile of the cells cultured in the presence of cholesterol and the cells cultured in broth alone when analyzing the effects of stressors on LAB. In contrast, when bile acids were added to the culture medium, no such similarity was observed [52]. Kimoto et al. [21] also noted changes in the distribution of fatty acids with *L. lactis* cells growing in the presence or absence of cholesterol, which were the result of its removal from the culture medium and its uptake into the cell membrane. These results could aid in understanding the effects of cholesterol on the metabolism of LAB and their impact on the quality of dairy products such as yogurt and kefir. Further research in this area could provide a better understanding of these effects and identify possible benefits or risks to human health.

## 4. Conclusions

In this study, it was found that the removal of cholesterol by lactobacilli cells, under gastric and intestinal juice conditions, was dependent on the biomass and viability of the bacterial cells and the bacterial strain studied. The binding of certain amounts of cholesterol by cells of LAB appears to be so strong that it is not released during its passage through the digestive system. The presence of cholesterol can affect the fatty acid profile of lactobacilli cells, which determines the structure and properties of the cell membrane and, in turn, can impact the metabolism and function of these bacteria. These findings may aid in understanding the effect of cholesterol on the metabolism of lactobacilli. However, it was observed that there was no significant effect of cholesterol supplementation on the survival of lactobacilli under gastric or intestinal juice conditions. Further research in this area could provide a better understanding of this effect and identify possible benefits or risks to human health.

Conversely, there was no significant effect of storage time, passage, or type of lactobacilli culture on the cholesterol content of fermented dairy products. This study also revealed differences in the cell survival rates of individual lactobacilli strains under gastric or intestinal juice conditions, depending on the environment in which the bacteria entered the tested system. During passage through the simulated stomach and intestines, the dry weight of products, such as significant milk protein content, had a more protective effect on lactobacilli bacterial cells than the fat content.

## Figures and Tables

**Figure 1 microorganisms-11-01478-f001:**
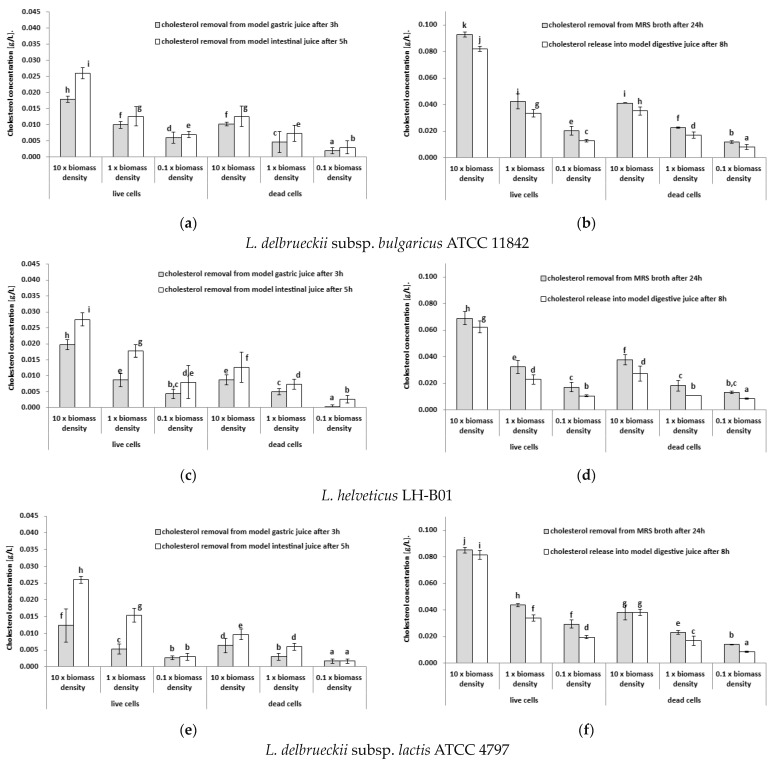
(**a**,**c**,**e**,**g**) Cholesterol removal using biomass samples with different concentrations of live and dead cells of lactobacilli strains under the gastric juice condition after 3 h or under the intestinal juice condition after 5 h. (**b**,**d**,**f**,**h**) Cholesterol removal or release using biomass samples with different concentrations of live and dead cells of lactic acid bacterial strains under the digestive juice condition after 8 h or in MRS broth after 24 h (mean values and SD values); ^a–j^ means with different letters in the same figure are significantly different (*p* < 0.05).

**Figure 2 microorganisms-11-01478-f002:**
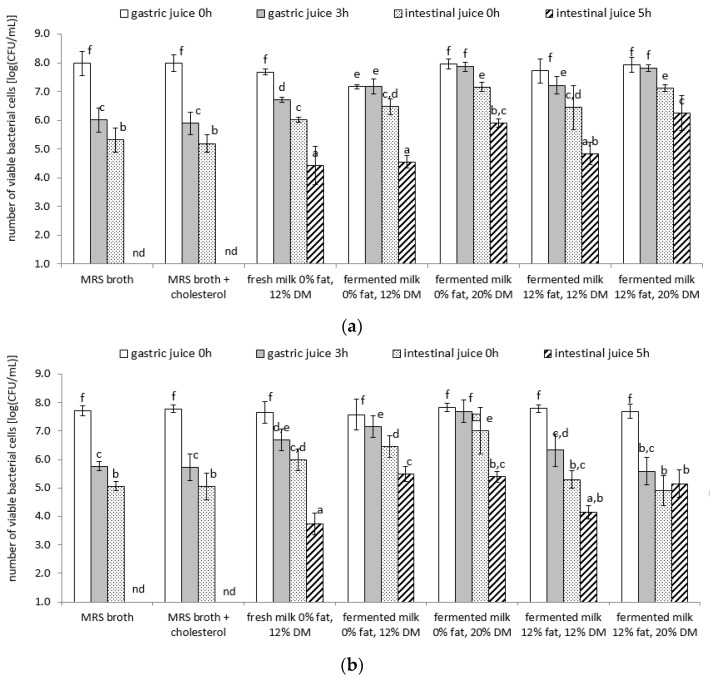
(**a**) *L. delbrueckii* subsp. *bulgaricus* ATCC 11842; (**b**) *L. helveticus* LH-B01; (**c**) *L. delbrueckii* subsp. *lactis* ATCC 4797; (**d**) *L. acidophilus* La-5. Survival of lactobacilli strains under digestive juice conditions (log(CFU/mL)); nd, not detected; ^a–f^ means with different lowercase letters in the same figure are significantly different (*p* < 0.05).

**Figure 3 microorganisms-11-01478-f003:**
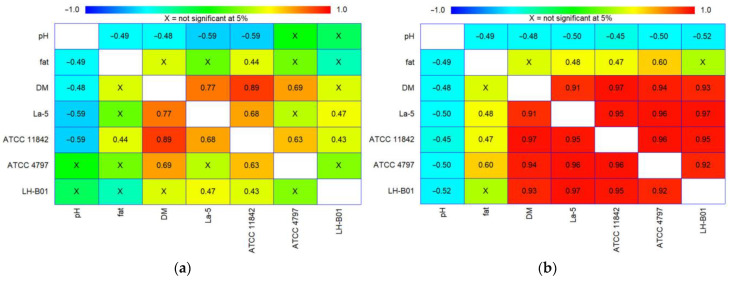
(**a**) gastric juice 3 h; (**b**) intestinal juice 5 h. Graphs of correlation matrix showing the relationship between pH, fat content, dry-matter content, and the viability of tested lactobacilli strains (*L. delbrueckii* subsp. *bulgaricus* ATCC 11842, *L. helveticus* LH-B01, *L. delbrueckii* subsp. *lactis* ATCC 4797, and *L. acidophilus* La-5) (**a**) after 3 h of incubation under gastric juice conditions, and (**b**) after 5 h under intestinal juice conditions (with a confidence level of 95.0%); x, not significant at 0.05.

**Table 1 microorganisms-11-01478-t001:** Changes in cholesterol content of different model product systems during cold storage and under digestive juice models (calculated as mg of cholesterol in 100 g of the primary product samples).

Strain	Product	Directly After Preparation	After 1 Week of Cold Storage	After 4 Weeks of Cold Storage	Gastric Juice Model after 3 h	Intestinal Juice Model after 5 h
*L. delbrueckii* subsp. *bulgaricus* ATCC 11842	MRS broth	0.0 ^a^ ± 0.00	0.0 ^a^ ± 0.00	0.0 ^a^ ± 0.00	0.0 ^a^ ± 0.00	0.0 ^a^ ± 0.00
	MRS broth + cholesterol	0.7 ^b^ ± 0.01	0.7 ^b^ ± 0.04	0.7 ^b^ ± 0.02	0.7 ^b^ ± 0.03	0.7 ^b^ ± 0.03
	Fresh milk 0.05% fat, 12% DM	1.7 ^c^ ± 0.04	1.6 ^c^ ± 0.02	1.6 ^c^ ± 0.04	1.7 ^c^ ± 0.05	1.6 ^c^ ± 0.05
	Fermented milk 0.05% fat, 12% DM	1.7 ^c^ ± 0.04	1.6 ^c^ ± 0.02	1.6 ^c^ ± 0.04	1.7 ^c^ ± 0.05	1.6 ^c^ ± 0.05
	Fermented milk 0.05% fat, 20% DM	1.6 ^c^ ± 0.04	1.5 ^c^ ± 0.02	1.5 ^c^ ± 0.04	1.6 ^c^ ± 0.05	1.5 ^c^ ± 0.05
	Fermented milk 12% fat, 12% DM	40.6 ^e^ ± 1.07	39.4 ^e^ ± 0.41	38.5 ^e^ ± 1.02	40.5 ^e^ ± 1.13	39.0 ^e^ ± 1.13
	Fermented milk 12% fat, 20% DM	37.5 ^d^ ± 0.99	36.5 ^d^ ± 0.38	35.6 ^d^ ± 0.95	37.4 ^d^ ± 0.97	36.0 ^d^ ± 0.97
*L. helveticus* LH-B01	MRS broth	0.0 ^a^ ± 0.00	0.0 ^a^ ± 0.00	0.0 ^a^ ± 0.00	0.0 ^a^ ± 0.00	0.0 ^a^ ± 0.00
	MRS broth + cholesterol	0.7 ^b^ ± 0.02	0.7 ^c^ ± 0.02	0.7 ^c^ ± 0.02	0.6 ^b^ ± 0.02	0.6 ^b^ ± 0.02
	Fresh milk 0.05% fat, 12% DM	1.6 ^c^ ± 0.04	1.6 ^c^ ± 0.06	1.6 ^c^ ± 0.01	1.6 ^c^ ± 0.05	1.6 ^c^ ± 0.05
	Fermented milk 0.05% fat, 12% DM	1.6 ^c^ ± 0.04	1.6 ^c^ ± 0.06	1.6 ^c^ ± 0.01	1.6 ^c^ ± 0.05	1.5 ^c^ ± 0.05
	Fermented milk 0.05% fat, 20% DM	1.5 ^c^ ± 0.04	1.5 ^c^ ± 0.05	1.5 ^c^ ± 0.01	1.5 ^c^ ± 0.04	1.4 ^c^ ± 0.04
	Fermented milk 12% fat, 12% DM	39.3 ^e^ ± 0.93	38.1 ^e^ ± 1.39	39.1 ^e^ ± 0.23	39.3 ^e^ ± 1.15	37.2 ^e^ ± 1.15
	Fermented milk 12% fat, 20% DM	36.4 ^d^ ± 0.85	35.3 ^d^ ± 1.28	36.2 ^d^ ± 0.22	36.4 ^d^ ± 1.09	34.5 ^d^ ± 1.09
*L. delbrueckii* subsp. *lactis* ATCC 4797	MRS broth	0.0 ^a^ ± 0.00	0.0 ^a^ ± 0.00	0.0 ^a^ ± 0.00	0.0 ^a^ ± 0.00	0.0 ^a^ ± 0.00
	MRS broth + cholesterol	0.6 ^b^ ± 0.03	0.7 ^b^ ± 0.01	0.7 ^b^ ± 0.02	0.6 ^b^ ± 0.01	0.6 ^b^ ± 0.01
	Fresh milk 0.05% fat, 12% DM	1.6 ^c^ ± 0.01	1.6 ^c^ ± 0.02	1.6 ^c^ ± 0.01	1.6 ^c^ ± 0.05	1.5 ^c^ ± 0.05
	Fermented milk 0.05% fat, 12% DM	1.6 ^c^ ± 0.01	1.6 ^c^ ± 0.02	1.6 ^c^ ± 0.01	1.6 ^c^ ± 0.05	1.5 ^c^ ± 0.04
	Fermented milk 0.05% fat, 20% DM	1.5 ^c^ ± 0.01	1.5 ^c^ ± 0.02	1.5 ^c^ ± 0.01	1.5 ^c^ ± 0.04	1.4 ^c^ ± 0.04
	Fermented milk 12% fat, 12% DM	37.9 ^e^ ± 0.29	38.6 ^e^ ± 0.56	38.0 ^e^ ± 0.25	37.9 ^e^ ± 1.01	36.5 ^e^ ± 1.01
	Fermented milk 12% fat, 20% DM	35.1 ^d^ ± 0.27	35.8 ^d^ ± 0.51	35.1 ^d^ ± 0.24	35.1 ^d^ ± 1.02	33.8 ^d^ ± 1.02
*L. acidophilus* La-5	MRS broth	0.0 ^a^ ± 0.00	0.0 ^a^ ± 0.00	0.0 ^a^ ± 0.00	0.0 ^a^ ± 0.00	0.0 ^a^ ± 0.00
	MRS broth + cholesterol	0.6 ^b^ ± 0.03	0.7 ^b^ ± 0.01	0.7 ^b^ ± 0.03	0.6 ^b^ ± 0.02	0.6 ^b^ ± 0.02
	Fresh milk 0.05% fat, 12% DM	1.6 ^c^ ± 0.06	1.6 ^c^ ± 0.03	1.6 ^c^ ± 0.02	1.6 ^c^ ± 0.04	1.5 ^c^ ± 0.05
	Fermented milk 0.05% fat, 12% DM	1.6 ^c^ ± 0.06	1.6 ^c^ ± 0.03	1.6 ^c^ ± 0.02	1.6 ^c^ ± 0.05	1.5 ^c^ ± 0.06
	Fermented milk 0.05% fat, 20% DM	1.5 ^c^ ± 0.05	1.5 ^c^ ± 0.02	1.5 ^c^ ± 0.02	1.5 ^c^ ± 0.05	1.4 ^c^ ± 0.05
	Fermented milk 12% fat, 12% DM	38.6 ^e^ ± 1.32	38.8 ^e^ ± 0.64	38.5 ^e^ ± 0.51	38.6 ^e^ ± 1.05	36.6 ^e^ ± 1.05
	Fermented milk 12% fat, 20% DM	35.8 ^d^ ± 1.23	35.9 ^d^ ± 0.59	35.7 ^d^ ± 0.48	35.7 ^d^ ± 1.07	33.9 ^d^ ± 1.07

^a,b,c,d,e^ Means with different lowercase letters within the entire table are significantly different (*p* < 0.05).

**Table 2 microorganisms-11-01478-t002:** Fatty acid composition (%, ×10^−3^) in the lactobacilli strain profiles obtained from the cells cultured in MRS broth with and without cholesterol (means ± SD).

Fatty Acid		*L. delbrueckii* Subsp. *bulgaricus* ATCC 11842		*L. helveticus* LH-B01		*L. delbrueckii* Subsp. *lactis* ATCC 4797		*L. acidophilus* La-5	
		MRS Broth	MRS Broth + Chol	MRS Broth	MRS Broth + Chol	MRS Broth	MRS Broth + Chol	MRS Broth	MRS Broth + Chol
C10:0	caproic/decanoic	0.9 ^a,b^ ± 0.6	0.2 ^a,b^ ± 0.2	0.3 ^a^ ± 0.1	0.5 ^a^ ± 0.2	0.5 ^a,b^ ± 0.2	1.5 ^a,b^ ± 0.9	1.7 ^b^ ± 0.6	0.8 ^b^ ± 0.2
C12:0	lauric/dodecanoic	4.7 ± 1.9	1.2 ± 0.0	11.8 ± 2.5	26.9 ± 1.1	4.4 ± 0.3	13.1 ± 2.1	10.0 ± 1.8	5.1 ± 0.1
C14:0	myristic/tetradecanoic	8.1 ± 2.0	4.1 ± 0.2	3.8 ± 1.7	3.2 ± 0.5	2.6 ± 0.6	7.6 ± 1.9	28.5 ± 2.9	15.6 ± 2.5
15:0, *iso*	*iso*-13-methyltetradecanoic	3.6 ^a,b^ ± 0.12	0.9 ^a,b^ ± 0.1	11.8 ^b^ ± 2.4	26.7 ^b^ ± 4.0	0.1 ^a^ ± 0.0	0.2 ^a^ ± 0.1	0.2 ^a^ ± 0.1	0.1 ^a^ ± 0.0
15:0, *anteiso*	*anteiso*-12-methyltetradecanoic	0.3 ± 0.2	0.1 ± 0.1	0.6 ± 0.9	1.4 ± 1.2	0.2 ± 0.1	0.7 ± 0.4	0.8 ± 0.3	0.4 ± 0.1
C15:0	pentadecanoic	0.1 ± 0.1	0.0 ± 0.0	0.5 ± 0.7	1.1 ± 1.0	0.1 ± 0.0	0.4 ± 0.2	0.4 ± 0.2	0.2 ± 0.0
C16:0	palmitic/hexadecanoic	24.3 ± 4.9	21.0 ± 6.8	14.8 ± 3.8	18.2 ± 2.7	14.2 ± 1.3	36.0 ± 1.6	091.3 ± 3.7	048.3 ± 5.0
C16:1, *trans*-9	palmitelaidic/*trans*-9-hexadecenoic	20.0 ^b^ ± 2.3	19.4 ^b^ ± 8.6	11.2 ^b^ ± 2.5	25.3 ^b^ ± 1.8	0.2 ^a^ ± 0.0	0.5 ^a^ ± 0.3	0.5 ^a^ ± 0.2	0.3 ^a^ ± 0.1
C16:1, *cis*-9	palmitoleic/*cis*-9-hexadecenoic	95.4 ^b^ ± 18.5	3.1 ^b^ ± 0.0	3.1 ^b^ ± 1.3	0.0 ^a^ ± 0.0	0.6 ^a^ ± 0.1	0.0 ^a^ ± 0.0	6.5 ^b^ ± 1.2	4.4 ^b^ ± 0.4
C12:0, 2OH	2-hydroxydodecanoic	0.3 ± 0.2	0.1 ± 0.1	0.8 ± 0.2	1.8 ± 0.6	0.1 ± 0.0	0.4 ± 0.2	0.5 ± 0.2	0.3 ± 0.1
*cycC17*:0, *cis*-9, 10	*cis*-9,10-methylenehexadecanoic	4.9 ^b^ ± 1.0	4.8 ^b^ ± 1.6	0.0 ^a^ ± 0.0	0.0 ^a^ ± 0.0	0.1 ^a^ ± 0.0	0.3 ^a^ ± 0.2	0.0 ^a^ ± 0.0	0.0 ^a^ ± 0.0
C18:0	stearic/octadecanoic	3.0 ± 0.8	5.4 ± 1.0	2.9 ± 0.5	6.7 ± 1.4	2.4 ± 0.4	8.2 ± 2.7	29.6 ± 2.5	13.8 ± 1.4
C18:1	octadecenoic	1.0 ± 0.6	0.3 ± 0.2	0.9 ± 0.4	2.1 ± 0.9	0.0 ± 0.0	0.0 ± 00	01.3 ± 0.5	0.6 ± 0.1
C18:1, *trans*-6	petroselaidic/*trans*-6-octadecenoic	0.0 ^a^ ± 0.0	0.0 ^a^ ± 0.0	0.7 ^b^ ± 1.0	1.5 ^b^ ± 0.4	0.0 ^a^ ± 0.0	0.0 ^a^ ± 0.0	0.1 ^a^ ± 0.0	0.0 ^a^ ± 0.0
C18:1, *trans*-9	elaidic/*trans*-9-octa-decenoic	0.2 ± 0.1	0.5 ± 0.4	0.0 ± 0.1	1.8 ± 0.8	0.0 ± 0.0	0.0 ± 0.0	0.0 ± 0.0	0.0 ± 0.0
C18:1, *trans*-11	trans-vaccenic/*trans*-11-octadecenoic	1.2 ± 0.7	0.3 ± 0.3	2.0 ± 1.9	4.4 ± 1.0	0.7 ± 0.2	2.0 ± 0.3	2.4 ± 0.9	1.2 ± 0.2
C18:1, *cis*-6	petroselinic/*cis*-6-octadecenoic	0.4 ^a^ ± 0.3	0.5 ^a^ ± 0.5	6.3 ^b^ ± 1.4	14.4 ^b^ ± 2.9	0.0 ^a^ ± 0.0	0.0 ^a^ ± 0.0	0.4 ^a^ ± 0.2	0.2 ^a^ ± 0.0
C18:1, *cis*-9	oleic/*cis*-9-octadecenoic	132.4 ^a,b^ ± 1.1	62.4 ^a^ ± 0.0	32.7 ^a^ ± 2.2	58.4 ^a^ ± 1.6	54.5 ^a^ ± 0.7	114.2 ^a,b^ ± 7.2	127.0 ^b^ ± 4.7	62.5 ^a^ ± 5.9
C18:1, *cis*-11	*cis*-vaccenic/*cis*-11-octadecenoic	17.9 ± 1.0	160.0 ± 4.7	57.4 ± 4.9	130.2 ± 7.1	2.5 ± 0.0	7.5 ± 0.2	5.9 ± 1.3	3.0 ± 0.6
C18:2, *trans*-9,*trans*-12	linoelaidic /*trans*-9,*trans*-12-octadecadienoic	0.8 ^b^ ± 0.5	0.2 ^a,b^ ± 0.2	0.0 ^a^ ± 0.0	0.0 ^a^ ± 0.0	0. ^a^ ± 0.0	0.1 ^a^ ± 0.1	0.0 ^a^ ± 0.0	0.0 ^a^ ± 0.0
C18:2, *cis*-9,*cis*-12	linoleic/*cis*-9,*cis*12-octadecadienoic	0.5 ^a^ ± 0.3	18.4 ^b^ ± 0.0	0.0 ^a^ ± 0.0	3.8 ^a,b^ ± 1.9	0.0 ^a^ ± 0.0	8.0 ^b^ ± 2.6	4.3 ^a,b^ ± 1.0	0.0 ^a^ ± 0.0
*cycC19*:0, *cis*-9, 10	dihydrosterculic /*cis*-9,10-methyleneoctadecanoic	63.2 ^b^ ± 8.7	10.8 ^b^ ± 0.0	7.0 ^a^ ± 1.5	0.0 ^a^ ± 0.0	7.8 ^a^ ± 1.2	6.8 ^a^ ± 0.7	41.3 ^b^ ± 2.0	5.3 ^a^ ± 1.5
*cycC19*:0, *cis*-10, 11	lactobacillic /*cis*-11,12-methyleneoctadecanoic	8.7 ^a^ ± 1.3	2.2 ^a^ ± 0.9	60.0 ^b^ ± 1.7	136.1 ^b^ ± 2.3	0.0 ^a^ ± 0.0	0.0 ^a^ ± 00	1.8 ^a^ ± 0.7	0.9 ^a^ ± 0.2
18:2, *cis*-9,*trans*-11	conjugated octadecadienoic	5.8 ± 1.5	1.5 ± 0.3	6.9 ± 0.2	15.6 ± 2.1	1.5 ± 0.5	4.5 ± 1.8	5.7 ± 1.2	2.9 ± 0.6
C18:2, CLA_1	conjugated octadecadienoic	0.5 ± 0.3	0.1 ± 0.1	0.7 ± 1.0	1.5 ± 0.4	0.2 ± 0.1	0.7 ± 0.4	0.5 ± 0.2	0.3 ± 0.0
18:2, *trans*-10,*cis*-12	conjugated octadecadienoic	3.1 ± 1.9	0.8 ± 0.7	7.4 ± 0.9	16.7 ± 1.1	2.3 ± 0.7	7.0 ± 2.3	6.2 ± 1.4	3.2 ± 0.6
C18:2, CLA_2	conjugated octadecadienoic	0.4 ± 0.2	0.1 ± 0.0	0.7 ± 0.1	1.5 ± 0.4	0.2 ± 0.1	0.6 ± 0.2	0.6 ± 0.2	0.3 ± 0.1
C18:2, CLA_3	conjugated octadecadienoic	0.1 ± 0.0	0.0 ± 0.0	0.6 ± 0.9	1.4 ± 0.2	0.2 ± 0.1	0.5 ± 0.3	0.6 ± 0.2	0.3 ± 0.1
C18:2, CLA_4	conjugated octadecadienoic	3.4 ± 1.1	0.9 ± 0.8	6.7 ± 1.8	15.1 ± 3.6	1.6 ± 0.5	4.9 ± 1.0	4.9 ± 0.9	2.5 ± 0.5

^a,b^ Means with different uppercase letters in the same row are significantly different (*p* < 0.05). For other fatty acids, there are no statistically significant changes.

**Table 3 microorganisms-11-01478-t003:** Composition of the external standard (bacterial acid methyl ester (BAME)) and identification parameters for each fatty acid.

Fatty Acid	Acid Name	RT [Min] *	ECL	EI *
C11:0	undecanoic	15.466	11.000	74, 87, 143, 157, 200
C12:0	lauric/dodecanoic	17.655	12.000	74, 87, 143, 214
C13:0	tridecanoic	19.751	13.000	74, 87, 143, 185, 228
C14:0	myristic/tetradecanoic	21.757	14.000	74, 87, 143, 199, 242
C10:0, 2OH	2-hydroxydecanoic	22.658	14.481	69, 83, 143, 228
C15:0, *iso*	*iso*-13-methyltetradecanoic	22.777	14.542	74, 87, 143, 213, 256
C15:0, *anteiso*	*anteiso*-12-methyltetradecanoic	23.084	14.701	74, 87, 143, 213, 256
C15:0	pentadecanoic	23.673	15.000	74, 87, 143, 213, 256
C16:0, *iso*	*iso*-14-methylpentadecanoic	24.647	15.541	74, 87, 143, 227, 270
C16:0	palmitic/hexadecanoic	25.505	16.000	74, 87, 143, 227, 270
C17:0, *iso*	*iso*-15-methylhexadecanoic	26.442	16.541	74, 87, 143, 241, 284
C16:1, *cis*-9	palmitoleic/hexadecenoic	26.485	16.565	69, 83, 96, 152, 236
C12:0, 2OH	2-hydroxydodecanoic	26.636	16.653	69, 83, 97, 171, 230
C17:0	heptadecanoic	27.255	17.000	74, 87, 143, 241, 284
*cycC17*:0, *cis*-9, 10	*cis*-9,10-methylenehexadecanoic	27.965	17.432	69, 74, 83, 97, 250
C18:0	stearic/octadecanoic	28.934	18.000	74, 87, 143, 255, 298
C12:0, 3OH	3-hydroxydodecanoic	29.175	18.159	71, 74, 83, 103
C18:1, *trans*-9	elaidic/octadecenoic	29.492	18.351	69, 74, 83, 97, 123, 264
C18:1, *cis*-9	oleic/octadecenoic	29.705	18.486	69, 74, 83, 97, 123, 264
C14:0, 2OH	2-hydroxytetradecanoic	30.221	18.797	69, 83, 97, 199
C19:0	nonadecanoic	30.545	19.000	74, 87, 143, 312
C18:2, *cis*-9,*cis*-12	linoleic/*cis*-9,*cis12*-octadecadienoic	31.015	19.307	97, 81, 95, 123, 294
* cyc * C19:0, *cis*-9, 10	dihydrosterculic/*cis*-9,10-methylene-octadecanoic	31.122	19.376	69, 74, 83, 97, 123, 278
C20:0	eicosanic	32.086	20.000	74, 87, 143, 326
C14:0, 3-OH	3-hydroxytetradecanoic	32.592	20.318	71, 74, 103
C16:0, 2-OH	2-hydroxyhexadecanoic	33.475	20.864	69, 83, 97, 227

* EI, electron ionization; RT, retention time.

## Data Availability

The data that support the findings of this study are available from the corresponding author (M.Z.) upon reasonable request.

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
