# Peer review of "Exploring the Cholesterol-Modifying Abilities of Lactobacilli Cells in Digestive Models and Dairy Products"

_microorganisms, 2023, doi:10.3390/microorganisms11061478_

Round 1
Reviewer 1 Report
Please correct according to the attached file

Author Response
Author's Reply to the Review Report (Reviewer 1)
Reviewer: Please correct according to the attached file.
Authors: We wanted to take a moment to express our deepest gratitude for your invaluable effort in reviewing and providing corrections for our manuscript. Your expertise and attention to detail have truly enhanced the quality of the text, and we are incredibly grateful for your time and dedication.
Your insightful suggestions and corrections have been immensely valuable in refining the article. Your careful review has not only improved the clarity and coherence of the manuscript but also strengthened the overall message. We truly appreciate the thoroughness with which you approached the task and the constructive feedback you provided.
Your comments have been carefully analyzed and incorporated into our text. The text has been adapted to all your comments.
Please know that your contributions have made a significant difference, and we are sincerely thankful for your commitment to improving the manuscript. Your expertise and guidance have not only helped us in this particular work but will also have a lasting impact on our future writing endeavors.
Reviewer 2 Report
It was a pleasure to review this paper. A well-organized experiment with a few mistakes (I wouldn't say small, because not writing chemical formulas correctly or not using italics when writing names are big mistakes, but they can be corrected).

Author Response
Author's Reply to the Review Report (Reviewer 2)
Reviewer: It was a pleasure to review this paper. A well-organized experiment with a few mistakes (I wouldn't say small, because not writing chemical formulas correctly or not using italics when writing names are big mistakes, but they can be corrected).
Authors: We would like to express our sincere appreciation for taking the time to review our paper. Your positive feedback and constructive comments have been immensely valuable to us.
We are grateful for your acknowledgement of the well-organized experiment and for pointing out the mistakes we made. We fully understand the importance of correctly writing chemical formulas and utilizing italics when writing names. Your feedback has helped us realize the significance of these errors and the impact they can have on the clarity and accuracy of the paper.
We want to assure you that we have taken your suggestions to heart and have diligently addressed the issues you raised. We have carefully corrected the mistakes and made the necessary revisions to ensure the accuracy and professionalism of our work. Thank you once again for your contribution to our manuscript.

Reviewer 3 Report
Comments to Authors
This study showed that: a) there was no significant effect of cholesterol supplementation on the survival of lactobacilli in model gastric juice or model intestinal juice; b) differences in the cell survival rates of individual lactobacilli strains in model gastric juice or model intestinal juice, depending on the environment in which the bacteria entered the tested system.
Authors are kindly requested to emphasize the current concepts about these issues in the context of recent knowledge and the available literature. This articles should be quoted in the References list.
References
1. Probiotic lactobacilli attenuate oxysterols-induced alteration of intestinal epithelial cell monolayer permeability: Focus on tight junction modulation. Food Chem Toxicol. 2023;172:113558. doi:10.1016/j.fct.2022.113558.
2. Salutary attributes of probiotic human gut lactobacilli for gut health. Lett Appl Microbiol. 2023; 76 (2): ovad011. doi:10.1093/lambio/ovad011.
3. Evaluation of the health properties of lactobacilli isolated from an Iranian traditional dairy product. Lett Appl Microbiol. 2023; 76 (2): ovac058. doi:10.1093/lambio/ovac058.
Minor editing of English language required
Author Response
Author's Reply to the Review Report (Reviewer 3)
Reviewer: This study showed that: a) there was no significant effect of cholesterol supplementation on the survival of lactobacilli in model gastric juice or model intestinal juice; b) differences in the cell survival rates of individual lactobacilli strains in model gastric juice or model intestinal juice, depending on the environment in which the bacteria entered the tested system.
Authors are kindly requested to emphasize the current concepts about these issues in the context of recent knowledge and the available literature. This articles should be quoted in the References list.
References
- Probiotic lactobacilli attenuate oxysterols-induced alteration of intestinal epithelial cell monolayer permeability: Focus on tight junction modulation. Food Chem Toxicol. 2023;172:113558. doi:10.1016/j.fct.2022.113558.
- Salutary attributes of probiotic human gut lactobacilli for gut health. Lett Appl Microbiol. 2023; 76 (2): ovad011. doi:10.1093/lambio/ovad011.
- Evaluation of the health properties of lactobacilli isolated from an Iranian traditional dairy product. Lett Appl Microbiol. 2023; 76 (2): ovac058. doi:10.1093/lambio/ovac058.
Author's Reply: We would like to express our heartfelt appreciation for your meticulous review of our manuscript. Your insightful comments and suggestions have greatly contributed to improving the quality and relevance of our study.
We sincerely thank you for highlighting the key findings of our research, specifically the lack of a significant effect of cholesterol supplementation on the survival of lactobacilli in both model gastric juice and model intestinal juice. Additionally, we appreciate your observation regarding the variations in cell survival rates among individual lactobacilli strains, depending on the environment in which they entered the tested system.
Your recommendation to emphasize the current concepts about these issues in the context of recent knowledge and the available literature is highly valued. We have taken your suggestion to heart and have carefully incorporated these relevant concepts into our revised manuscript. Moreover, we have ensured that the references you suggested have been included in our reference list, thereby strengthening the scientific foundation of our work.
Reviewer: Minor editing of English language required.
Author's Reply: We would like to express our gratitude for pointing out the need for minor editing of the English language. We have dedicated considerable effort to improving the language and clarity of our manuscript, addressing any grammatical or stylistic issues that may have affected its readability.
We are genuinely grateful for the time and effort you have invested in providing us with such valuable feedback.
Once again, we want to emphasize our sincere appreciation for your contributions to our study. We have diligently considered and incorporated all of your suggestions into our revised manuscript, as well as utilized the recommended references in our text. Thank you sincerely for your invaluable guidance and support.

Reviewer 4 Report
Ziarno et al., did a comprehensive study on the cholesterol-modifying ability of Lactobacilli cell in digestive models and dairy products. I am not an expert in experimental microbiology, but the methods, results, and discussion read sound to me. The paper is well written and of the interest to the general readership of the journal Microorganisms. Please see my comments below:
1. I think there are too many takeaway messages in the abstract. The authors may narrow the takeaway message into 3-4 points.
2. Line 33, disease is one word.
3. Line 43-44, citations?
Author Response
Author's Reply to the Review Report (Reviewer 4)
Review: Ziarno et al., did a comprehensive study on the cholesterol-modifying ability of Lactobacilli cell in digestive models and dairy products. I am not an expert in experimental microbiology, but the methods, results, and discussion read sound to me. The paper is well written and of the interest to the general readership of the journal Microorganisms. Please see my comments below:
Authors: Dear Reviewer, thank you sincerely for your valuable feedback and for taking the time to review our manuscript. We greatly appreciate your positive remarks regarding the comprehensive study conducted by us on the cholesterol-modifying ability of lactobacilli cells in digestive models and dairy products. Your acknowledgment of the paper being well-written and relevant to the general readership of the journal Microorganisms is encouraging.
We would like to express our gratitude for your thorough evaluation of our manuscript and for providing insightful comments. Your expertise in experimental microbiology is highly appreciated, and we have carefully considered all of your suggestions during the revision process. Your feedback has undoubtedly contributed to enhancing the clarity and strength of our research.
We are pleased to inform you that we have incorporated your comments into the revised manuscript, addressing each point specifically. We have made the necessary revisions to ensure the methods, results, and discussion are presented accurately and coherently.
Review: 1. I think there are too many takeaway messages in the abstract. The authors may narrow the takeaway message into 3-4 points.
- Line 33, disease is one word.
- Line 43-44, citations?
Authors: Thank you for your valuable feedback and suggestions regarding the abstract of our manuscript. We appreciate your input and have carefully considered your recommendations. We have made the necessary revisions to shorten the abstract while still conveying the key findings and significance of the study.
We hope that these revisions meet your expectations and address the need for a more concise abstract. Thank you once again for your valuable input and for your continued support throughout the review process.